# Metal–Organic Framework (MOF) through the Lens of Molecular Dynamics Simulation: Current Status and Future Perspective

**Amin Hamed Mashhadzadeh** [1,*], **Ali Taghizadeh** [1], **Mohsen Taghizadeh** [1], **Muhammad Tajammal Munir** [2,3], **Sajjad Habibzadeh** [4], **Azam Salmankhani** [5], **Florian J. Stadler** [6,*] **and Mohammad Reza Saeb** [7,*]

1   Center of Excellence in Electrochemistry, School of Chemistry, College of Science, University of Tehran, Tehran 14155-6455, Iran; ali_taghizadeh1995@yahoo.com (A.T.); taghizadeh.mohsen1995@gmail.com (M.T.)
2   College of Engineering and Technology, American University of the Middle East, Egaila 54200, Kuwait; tajammal.munir@auckland.ac.nz
3   Department of Chemical and Materials Engineering, The University of Auckland, Auckland, Private Bag 92019, New Zealand
4   Department of Chemical Engineering, Amirkabir University of Technology (Tehran Polytechnic), Tehran 15916-39675, Iran; sajjad.habibzadeh@aut.ac.ir
5   Faculty of Mechanical Engineering, K. N. Toosi University of Technology, Tehran 196976-4499, Iran; azamsalmankhani@gmail.com
6   College of Materials Science and Engineering, Shenzhen Key Laboratory of Polymer Science and Technology, Guangdong Research Center for Interfacial Engineering of Functional Materials, Nanshan District Key Laboratory for Biopolymers and Safety Evaluation, Shenzhen University, Shenzhen 518055, China
7   Department of Resin and Additives, Institute for Color Science and Technology, Tehran P.O. Box: 16765-654, Iran
*   Correspondence: amin.hamed.m@gmail.com (A.H.M.); fjstadler@szu.edu.cn (F.J.S.); saeb-mr@icrc.ac.ir (M.R.S.); Tel.: +98-9113765114 (A.H.M.); +86-0755-8671-3986 (F.J.S.); +98-9128264307 (M.R.S.)

**Abstract:** As hybrid porous structures with outstanding properties, metal–organic frameworks (MOFs) have entered into a large variety of industrial applications in recent years. As a result of their specific structure, that includes metal ions and organic linkers, MOFs have remarkable and tunable properties, such as a high specific surface area, excellent storage capacity, and surface modification possibility, making them appropriate for many industries like sensors, pharmacies, water treatment, energy storage, and ion transportation. Although the volume of experimental research on the properties and performance of MOFs has multiplied over a short period of time, exploring these structures from a theoretical perspective such as via molecular dynamics simulation (MD) requires a more in-depth focus. The ability to identify and demonstrate molecular interactions between MOFs and host materials in which they are incorporates is of prime importance in developing next generations of these hybrid structures. Therefore, in the present article, we have presented a brief overview of the different MOFs' properties and applications from the most recent MD-based studies and have provided a perspective on the future developments of MOFs from the MD viewpoint.

**Keywords:** metal–organic frameworks; diffusion; drug delivery; water treatment; molecular dynamics simulation

## 1. Introduction

Porous coordination polymers (PCPs), so-called metal–organic frameworks (MOFs), such as robust 1-3D inorganic/organic complexes with elevated chemical versatility, are made by the combination of single or mixed metal ions and bio-based or organic ligands as a bridge [1–3]. Figure 1 shows the components of commonly used MOFs. The angle and orientation between linkers and nodes identify the shape and size of pores in the MOF's lattice structure. Furthermore, the mechanical/chemical stability of the fabricated MOFs directly relies on the bond strength between the nodes and linkers. Moreover, the chemical activity of these complexes is related to the nature of the metal nodes and the activity of organic linkers [4].

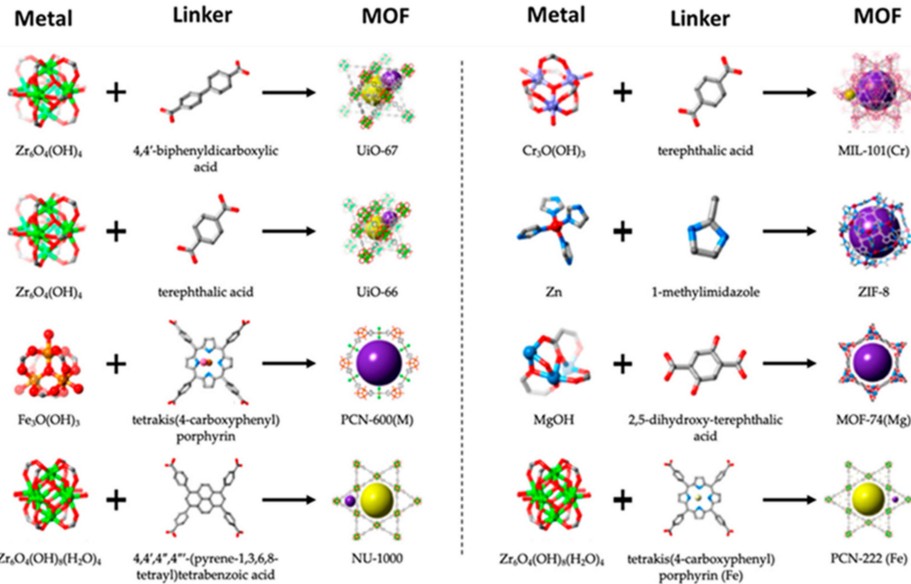

**Figure 1.** Schematic illustration of the molecular structure of common metals and ligands and the MOFs resulting from their combination.

According to the literature, the fabrication methods and synthesis conditions such as the temperature can alter the morphology and physicochemical properties of the MOFs [5,6]. Figure 2 introduces the various fabrication routes and possible reaction temperatures. The broad spectrum of options for selecting metal nodes and linkers, as well as the ability for various post-modifications, led to the development of hundreds of thousands of MOFs [4–6].

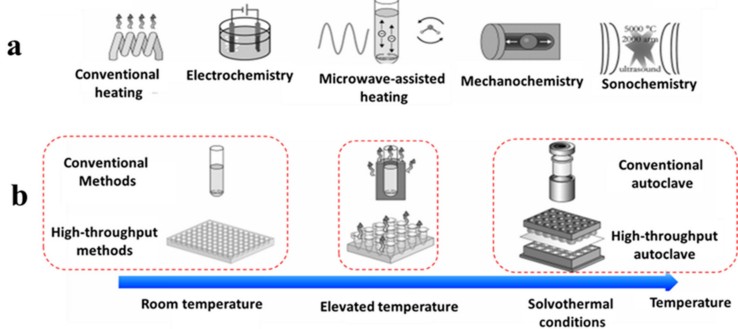

**Figure 2.** Illustration of various reported MOF fabrication techniques in the literature; conventional hydrothermal and microwave-assisted hydrothermal approaches are the most commonly applied approaches, while electrochemical, mechanochemistry, and ultrasound-assisted methods are quite new techniques in the synthesis of MOFs (**a**) [7]. A wide range of temperatures from room conditions to solvothermal conditions are applied in the fabrication of MOFs (**b**) [8].

For the last 20 years, due to their distinctive characteristics, such as a controllable pore size, permanent porosity, high chemical/mechanical stability, elevated active surface area, and facile functionalization, MOFs have been used as adsorbents [9,10], catalysts [11,12], supercapacitors [13], and drug delivery vehicles [14,15], especially for cancer therapy purposes [16,17]. Moreover, recently, sophisticated mixed-metal MOFs [18,19], as well as surface-modified MOFs [20], BioMOFs [21,22], and MOFs-derived materials [4,23], have attracted much interest in various fields, ranging from environmental engineering to biomedicine. The high number of publications related to MOFs, coupled with a sharp increase in the number of released papers each year, exhibits the desirability to use these valuable complexes (Figure 3).

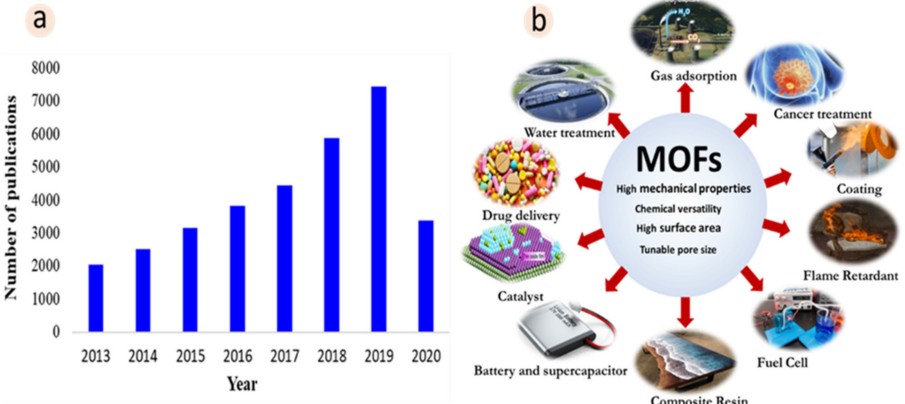

**Figure 3.** (**a**) The number of published articles related to MOFs in the period of 2013 to 2020 (Source: Scopus, April 28, 2020). (**b**) Some unique characteristics and various applications of MOFs (designed by the authors of the present work).

However, since the required solvents, linkers, and metal precursors, as well as conventional MOF fabrication methods such as ultrasound-assisted, microwave, or hydrothermal approaches, are quite expensive, and since, additionally, the synthesis process is time-consuming, research focused on exploring these structures from a theoretical viewpoint. Hence, molecular simulation approaches such as molecular dynamics simulations (MD) have been proposed as fast and low-cost alternatives for providing a comprehensive understanding of MOFs' capabilities, screening their behavior and comparing them to experimental data, while also predicting their further applications in a broader range of industries [24–29].

## 2. Adsorption/Diffusion Properties of MOFs

The outstanding properties of MOFs originate from their high internal surface area, as well as the flexibility of the organic linkers to swing [30,31], which enables them to let the guest molecules pass through their narrow gates (diffusion) [32]. This also makes them potential candidates for the adsorption of gaseous elements [33–35], removal of acids and heavy metals [36–39], and separation of gases and other contaminants from different environments [40–44]. In addition to experimental research, theoretical approaches, including molecular dynamics simulation, have been widely employed to investigate the diffusion and adsorption properties of MOFs. Molecular dynamics simulation (MD) was used by Chokbunpiam et al. to probe the diffusion selectivity of the $N_2/NO_2$ mixture in three ZIF materials, including ZIF-8, ZIF-90, and ZIF-78, with the study reporting a higher diffusion selectivity of ZIF 78>90>8 [45]. Wehbe et al. probed the possibility of removing lead atoms from water using UiO66-MOFs using MD simulations and the large-scale atomic/molecular massively parallel simulator (LAMMPS) package [36]. They considered the viability of adsorption of $Pb^{2+}$ ions onto the UiO66 (Figure 4) as a function of density particles, metal cation concentration, and the presence of $NO_3-$ ions and found that the favorable condition to achieve a higher lead adsorption volume occurred at a low number of density particles, an absence of $NO_3-$, and a high metal cation concentration.

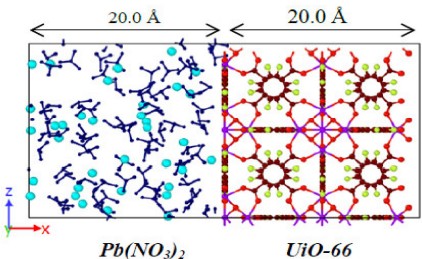

**Figure 4.** Snapshot of a simulation system at t = 0 ns, containing 37 $Pb^{2+}$ ions and 74 $NO_3^-$ ions. Color code: Maroon (C in UiO-66), Red (O in UiO-66), Purple (Zr in UiO-66), Green (H in UiO-66), Navy Blue ($NO_3^-$), and Cyan ($Pb^{2+}$). Water molecules are not shown in the figure for clarity (Ref. [36]).

Bigdeli et al. explored the adsorption and diffusion of Terephthalic acid (TPA) from water by different MOFs in a study based on MD simulations [46]. They modeled six types of MOFs, as displayed in Figure 5, and reported a higher gradient of the Mean Square Displacement (MSD), higher diffusion coefficient, and higher adsorption energy of MIL-101(Cr) amongst all studied MOFs, providing more possibility for TPAs to diffuse in this type of MOF or of being adsorbed by it.

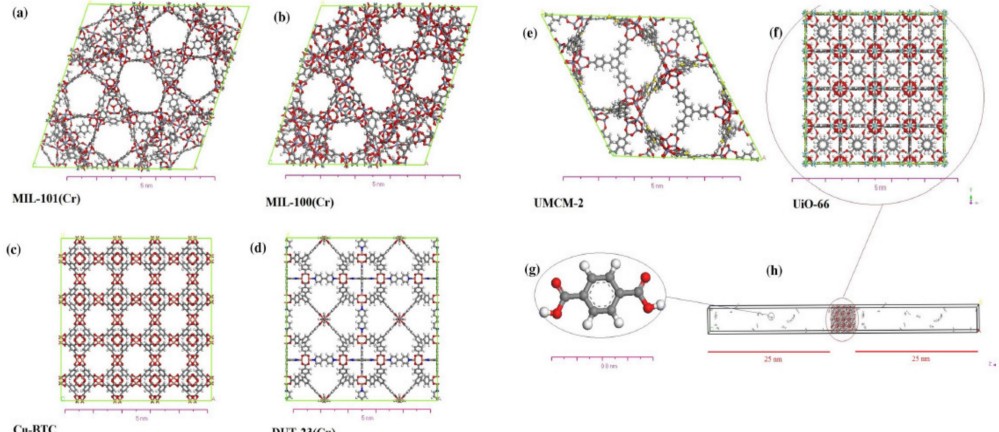

**Figure 5.** (**a**) MIL-101(Cr), (**b**) MIL-100(Cr), (**c**) Cu-BTC, (**d**) DUT-23(Cu), (**e**) UMCM-2, (**f**) UIO-66, (**g**) TPA molecule, and (**h**) simulation box (Ref. [46]).

In another MD case, Fan et al. employed ZIF-11 MOF to consider the diffusivity and adsorption of alcohol as well as hydrocarbon vapors with loading at 308 K and compared their results with those obtained for ZIF-8 [44]. They showed a higher diffusivity and lower activation energy of ZIF-11 compared to ZIF-8 and related it to the greater flexibility of the benzoimidazole linker in ZIF-11 compared to the 2-methylimidazole linker in ZIF-8, emphasizing the importance of flexibility in the diffusion properties of MOFs. They also investigated the diffusion of methane as a function of the temperature and found an inconsistent activation energy due to the dependence of the ligands' flexibility to the temperature. Furthermore, comparing the results of alcohol and hydrocarbon vapors revealed a slower diffusion of alcohol in ZIF-11 compared to hydrocarbon vapors. Ghoufi and Maurin modeled the diffusion of neo-pentane in the ID-channel of MIL-47(V) MOF. By analyzing the diffusion mechanism of neo-pentane along the *xy* and *yz* directions, they confirmed the abnormal diffusion process of neo-pentane regardless of loading, which had earlier been reported in an experimental study as well [47]. They also showed that this abnormal diffusion was not correlated with MIL-47(V) flexibility but was due to the size of the pores confining the movement of neo-pentane molecules inside the MOF channels. Besides the research mentioned above regarding MOFs' diffusion and adsorption properties under the MD framework, other MD-based achievements in this regard are presented in Table 1.



**Table 1.** Adsorption/diffusion properties of MOFs from a molecular dynamics perspective.

| No | MOF Type | Adsorbate | Main Findings | Ref. |
|---|---|---|---|---|
| 1 | ZIF-78 | $CH_4/CO_2$ | Same $N_2/O_2$ ratio in the gas phase and adsorbed phase. Decreasing the temperature and increasing the pressure increased selectivity. | [48] |
| 2 | ZIF-8 | $N_2/CO_2$ | Independent adsorption of $N_2$ to temperature unlike $CO_2$, * gate opening was found for $N_2/CO_2$ mixture at 20 molecules per cage, a substantial decrease of the diffusion coefficient at high loadings. | [49] |
| 3 | MIL-53(Al) | $CO_2$ | Significant increase in the enthalpy of adsorption with an increasing pressure, demonstrating that the adsorption of $CO_2$ in the MIL-53 structure was very sensitive to structural parameters. | [50] |
| 4 | ZIF-68 and ZIF-70 | $CH_4/H_2$ and $CH_4/CO_2$ | Diffusion of $CH_4$ increased with the concentration of $H_2$ in the $CH_4/H_2$ mixture, while it was independent of the $CO_2$ concentration in the $CH_4/CO_2$ mixture, whatever the MOF type. | [51] |
| 5 | Mg-MOF-74 MIL-101(Cr) UiO-66 ZIF-8 Ce-BTC | $H_2S$ in $H_2S/CO_2$ mixture | A complete reversible physical adsorption occurred. | [52] |
| 6 | ZIF-7 | $H_2$ | Decoration of MOF with Sc increased the binding energy and the number of $H_2$ adsorbed. | [53] |
| 7 | MIL-53(M) M=Cr, Fe, Sc, Al | $CH._4$, $N_2$, $CO_2$, $H_2S$ | Temperature-dependent adsorption capacity, independent of the cluster type, higher diffusion of $CH_4$ compared to other adsorbates. | [54] |
| 8 | ZIF-10 | $CH._4$, $SO_2$, $CO_2$ | Decrease in self-diffusion coefficient of $CH._4$ with the loading, while for $SO_2$ and $CO_2$ it increased at low uptakes and decreased at higher ones. | [55] |
| 9 | ZIF-8 | $C_2H_6$, $C_3H_8$ | Slower diffusion of $C_3H_8$ compared to $C_2H_6$. Flexibility of the MOF framework and gate opening phenomena facilitated adsorption and diffusion. | [56] |
| 10 | HKUST-1, CuBDC(ted)0.5 Zn-MOF-74 MIL-100(Fe) MOF-5 | $H_2S$ in $H_2S/CO_2$ mixture | Disposable chemical reaction was found. | [52] |

* The change in the lattice shape of the MOF's structure opens bottlenecks, allowing larger molecules to enter and pass through its cavities simply; this phenomenon is called gate opening.

## 3. MOFs for Water Desalination

Following the population explosion over the past few decades, the growing demands for freshwater around the world resulted in severe limitations of water supplies. Eventually, providing drinkable water from other sources such as saline water has become of high importance. Reverse Osmosis (RO) membranes have been used as a solution for extracting fresh water from saline water [57,58].

However, using the conventional RO membranes on a large scale would be costly and highly energy-consuming [59]. MOFs, especially those composed of organic linkers with a higher acidity and metals with a higher valence charge, have recently been introduced as promising RO membranes due to their high stability in an aqueous environment owing to the strong bonds that exist between the metals and linkers [60].

In this regard, various MOFs such as UiO-66 [61], CPO-27Ni [62], aluminum fumarate [63], and MIL-101(Cr) [64] have been probed in experimental studies as appropriate desalination membranes. However, exploring MOFs as RO membranes using theoretical methods such as molecular dynamics is a very recent development, and the number of articles on this subject is almost limited. Molecular dynamics simulation was used by Jeffery et al. to identify the interactions of water with MOF-5 [65] for the first time, and it showed that MOF-5 was very stable at low water percentages while it was unstable when the water content was over 4% (Figure 6).

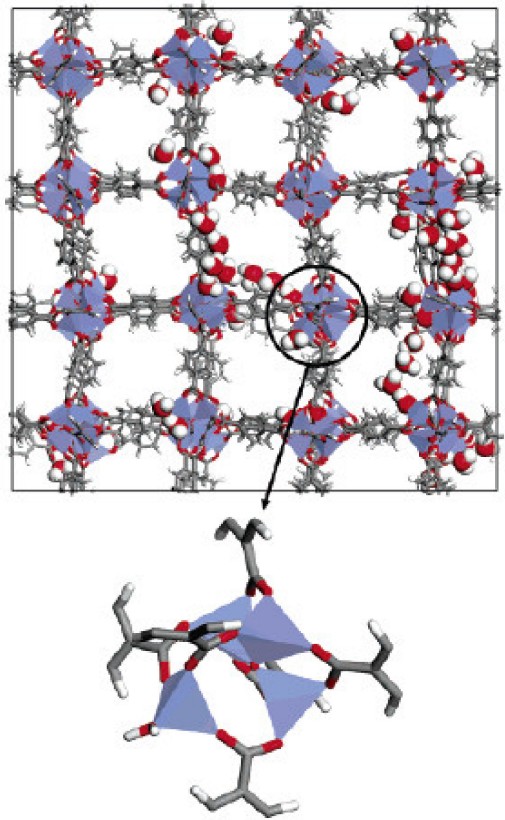

**Figure 6.** Disruption of MOF-5 in water (Ref. [65]).

Using MD simulations, Lyu et al. investigated the effect of relevant material defects on the desalination properties of pristine and defective UiO-66 MOFs [66]. They considered the water permeability and salt rejection rate of defect-free MOF, MOF with one missed-linker, and MOF with one missed-cluster. They showed that MOF with a cluster-missed defect had a water permeability of 800 L m-2 h-1 bar-1, which is far higher than that obtained for defect-free MOF (50 L m-2 h-1 bar-1). This defective MOF also exhibited the highest salt rejection percentage (at >99%) among all studied cases. They also mentioned that introducing hydrophobic compensation groups to defective MOFs could enhance the water intrusion, so the needed pressure for membrane saturation could be decreased more than ten times. Using molecular dynamics simulation, Cao et al. simulated a membrane system package including Ni/Cu-HAB MOFs with 1–3 layers, graphene, and MoS$_2$ membranes, a saline water box (contains potassium and chloride at a molarity of ~2.28 M), a graphene piston to apply pressure on saline water, and a freshwater box [67].

They observed that at a pressure of 100 bar the water fluxing through single-layer MOF was nine times higher than that of graphene or $MoS_2$. MOF membranes with two layers could reject nearly 100% of the ions. Furthermore, they reported that the water permeation rate of their suggested MOF membrane was between 3–6 times higher than the rates of common commercial membranes like MFI zeolite or brackish RO. Dahanayaka et al. investigated the desalination performance of a graphene oxide (GO)/HKUST-1 MOF composite membrane under the MD framework for pervaporation (PV) membrane fabrication [68]. They demonstrated that adding GO could decrease the water affinity of Cu atoms and improve the KHUST-1 stability in a water-based environment. However, increasing the number of barriers in the molecular paths as a result of adding GO resulted in a water flux reduction.

## 4. Drug Storage and Drug Delivery

In recent years, considerable attention has been devoted to the biomedical applications of MOFs such as biosensing [69,70], bioimaging [71], and pharmaceutical applications like drug storage [72] and drug delivery [73]. Since the pioneer drug delivery vehicles, including polymers, peptides, metal clusters, and carbon-based nanostructures, suffer from significant drawbacks in terms of toxicity, a high release speed, and a limited storage capacity [74], MOFs are proposed as potential alternatives to provide a progressive [75], controllable [76], and non-toxic [77] delivery due to their highly tunable properties (linkers and metals) as well as to their porosity.

Most of the research over the past decade regarding the role of MOFs in drug delivery applications has been conducted from an experimental point of view, while the theoretical investigation of MOFs from a drug delivery perspective is more novel. Molecular dynamics simulation provides a good insight into the storage and release mechanisms in MOFs, enabling us to identify the favorable sites for drug hosting, as well as to define the molecular interactions between the drug and the MOF host. Using MD simulation, it is possible to consider intermolecular and intramolecular interactions to calculate the forces, positions, and velocity of the drug molecules (based on Newton's second law) once the equilibrium is reached, and consequently to compute the diffusion coefficients of the drug molecules in MOFs [78,79]. In an experimental–theoretical study, Zr-based MOFs, including UiO-66 and UiO-67 coated with modified poly(ε-caprolactone), were studied by Fillipousi et al. as favorable anticancer carriers for cisplatin, and MD was used in order to provide a good visualization of the favorable adsorption sites of the adsorbates in MOFs [80]. Figure 7 shows the MD-provided snapshots of the adsorption of cisplatin molecules in UiO-66 and UiO-67 for t = 5000 fs.

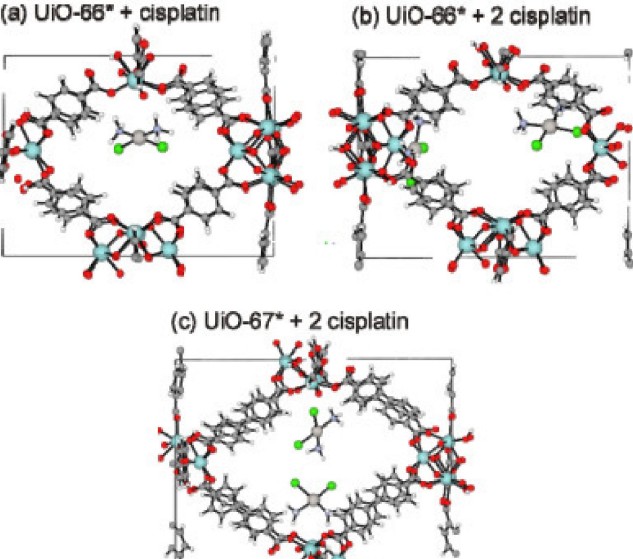

**Figure 7.** Snapshots of cisplatin stored in UiO-66 and UiO-67 coated with modified poly(ε-caprolactone) based on MD simulations provided by Fillipousi et al. (Ref. [80]).

Eura et al., in their MD-based work, compared the drug and cosmetic storage efficiency of MIL-100(Fe), MIL-101(Cr), MIL-53(Fe), zeolites, and mesoporous silica (MCM-41) [72]. They reported a good agreement between their results and those obtained by experimental studies regarding the storage and release of caffeine, urea, and ibuprofen. They also found a slow diffusion of drug molecules in MOFs, confirming the controllable and efficient delivery by MOFs compared to traditional drug vehicles. In another MD case, Eura et al. extended their work to investigate the drug delivery capacity of MOFs for delivering anti-cancer drugs [81]. By selecting MOF-74 as a carrier of two anti-cancer drugs, including methotrexate (MTX) and 5-fluorouracil (5-FU) (as shown in Figure 8), they found stronger interactions and a tighter adsorption between MTX and MOF-74 at a lower fugacity, whereas at higher ones 5-FU showed a better adsorption due to higher entropic effects.

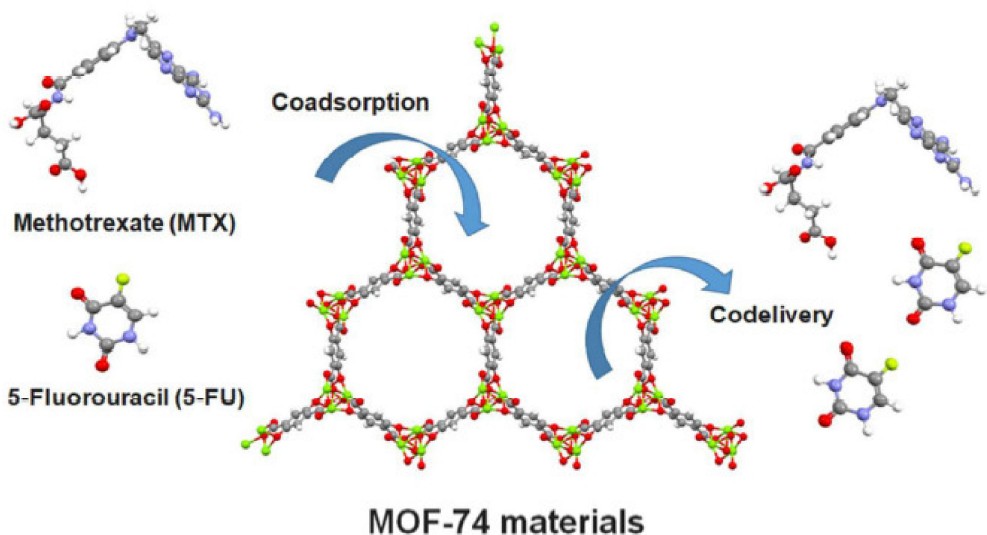

**Figure 8.** MD simulation of MOF-74 for the storage and delivery of anti-cancer drugs (Ref. [81]).

IRMOF-74-III was considered by Kotzabasaki et al. as a potent storage for Gemcitabine (GEM) delivery [82]. Using molecular dynamics simulation, they reported a slow diffusion of GEM inside the IRMOF-74-III, demonstrating a controlled drug release as a crucial factor for drug delivery applications. In another MD-based article, Shahabi and Raissi investigated the drug delivery performance of peptide-based MOFs (MPF) for 6-mercaptopurine (6-MP) as a function of an external electric field [83]. They showed that the drug molecules had stronger interactions with MPF at lower electric field intensities when compared to higher ones, so applying the electric field (EF) did not affect the drug storage efficiency positively. Increasing the electric field strength resulted in higher dynamic movements and a lower diffusion coefficient, emphasizing the adverse effect of EF on intermolecular interactions.

## 5. Other MOFs Applications from a Molecular Dynamics Perspective

Other applications and properties of these hybrid crystalline porous materials, far from the abovementioned properties and performances of MOFs, such as ion transportation and ion conduction [84,85], proton transportation [86,87], catalytic performance [88,89], and energy storage [90–92], have also been investigated with an MD approach on a small scale.

## 6. Conclusions

Relying on the versatile and tunable properties of MOFs, including a high specific surface area, selective adsorption/diffusion, low density, and high diversity, MOFs are assumed to be hybrid crystalline porous structures potentially appropriate for a wide variety of industrial applications. Those properties and performances of MOFs that have so far been explored by molecular dynamics (MD) simulation are in close agreement with experimental reports. Therefore, there is a great opportunity

for the prediction and development of new MOF-including structures, such as polymer/MOFs nanocomposites and MOFs heterostructures, using MD. It is believed that this theoretical approach could be employed on a larger scale for the further exploration of MOFs and could extend their industrial applications in the near future. Taking a molecular look at the interfacial interactions between the MOF and the host material give rise to deepening our understanding of the performance of MOF; thereby, it would be possible to develop tailor-made MOF for higher performance applications.

**Author Contributions:** A.H.M. and M.R.S. did conceptualization; M.T., A.T. & A.S. generated formal analysis; Investigations and data curation were managed by M.T.M. & S.H.; writing—original draft done by A.H.M., A.S., A.T. & M.T.; writing—review and editing by F.J.S. & M.R.S. to give the review its final form; visualization and final checking of the whole manuscript performed by M.R.S. All authors have read and agreed to the published version of the manuscript.

**Funding:** This research received no external funding.

**Conflicts of Interest:** The authors declare no conflict of interest.

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
