# Peer review of "Metal–Organic Framework (MOF) through the Lens of Molecular Dynamics Simulation: Current Status and Future Perspective"

_jcs, doi:10.3390/jcs4020075_

Round 1
Reviewer 1 Report
I feel reluctant to recommend this work, albeit an intriguing topic, for publication in JSC before the following concerns are addressed.
(1) Has any author contribute to this specific field (MOF & MD)? A statement like "we......" was made on Page 7 (last paragraph), but neither sufficient details nor appropriate references were provided.
(2) This work is resulted from a collaboration among several researchers all over the world. It would be helpful if their contributions are clearly stated.
(3) Many figures in this paper are copied from literature. Have the authors gotten permission to use them from a copyright perspective?
(4) Simply mistakes can be found throughout this paper, and these should be avoided whatsoever.
Author Response
Authors’ Response to the Review Comments
Journal: Journal of Composite Science
Manuscript #: jcs-831696
Title of Paper: Metal-organic Framework (MOF) through the Lens of Molecular Dynamics Simulation: Current Status and Future Perspective
Authors: Amin Hamed Mashhadzadeh, Ali Taghizadeh, Mohsen Taghizadeh, Muhammad Tajammal Munir, Sajjad Habibzadeh, Azam Salmankhani, Florian J. Stadler, Mohammad Reza Saeb
Dear Editor,
We appreciate the time and efforts spent by the reviewers in qualification of this manuscript. The manuscript has been carefully revised according to the useful comments and suggestions of the reviewers. A point-by-point response to the reviewers’ comments has also been provided. In the revised manuscript we used different colours so as to have attention of reviewers making them able to easily track changes applied to the text. We hope that the revised manuscript can lure the attention of reviewers and respected editor as well.
Thanks for your kind consideration in advance.
Yours sincerely,
Dr. Amin Hamed Mashhadzadeh
Prof. Florian J. Stadler
Dr. Mohammad Reza Saeb
Reviewers' comments:
Reviewer #1
I feel reluctant to recommend this work, albeit an intriguing topic, for publication in JSC before the following concerns are addressed.
- Has any author contribute to this specific field (MOF & MD)? A statement like "we......" was made on Page 7 (last paragraph), but neither sufficient details nor appropriate references were provided.
Response:
- Thanks for your kind attention. In that specific sentence authors ment that “it is possible to consider” not “we consider”. This unwanted mistake has been corrected. Besides, we have submitted several papers in the field of MOF and MD individually. Currently, because of the pandemic the review process in almost every single journal has been prolonged, and some of the submissions related to MOFs and MD have remained unpublished. However, we have several more manuscripts under review and also some published in different peer-reviewed journals related to MOFs and MD simulation.
Authors’ papers related to MOFs:
[1] Synthesis of pearl necklace-like ZIF-8@chitosan/PVA nanofiber with synergistic effect for recycling aqueous dye removal
[2] In situ deposition of Ag/AgCl on the surface of magnetic metal-organic framework nanocomposite and its application for the visible-light photocatalytic degradation of Rhodamine dye
[3] Novel magnetic amine functionalized carbon nanotube/metal-organic framework nanocomposites: From green ultrasound-assisted synthesis to detailed selective pollutant removal modelling from binary systems
[4] Activated carbon/metal-organic framework nanocomposite: Preparation and photocatalytic dye degradation mathematical modeling from wastewater by least squares support vector machine
[5] Ultrasound-assisted green synthesis and application of recyclable nanoporous chromium-based metal-organic framework
[6] Activated carbon/metal-organic framework composite as a bio-based novel green adsorbent: Preparation and mathematical pollutant removal modelling
[7] Synthesis, characterization, and high potential of 3D metal–organic framework (MOF) nanoparticles for curing with epoxy
Authors’ papers related to MD:
[1] Molecular dynamic study of mechanical properties of single/double wall SiCNTs: Consideration temperature, diameter and interlayer distance
[2] Mechanical properties of silicon-germanium nanotubes: A molecular dynamics study
[3] Effect of point defects and low-density carbon-doped on mechanical properties of BNNTs: A molecular dynamics study
[4] Density functional theory based molecular dynamics study on hydrogen storage capacity of C24, B12N12, Al12 N12, Be12O12, Mg12O12, and Zn12O12 nanocages
[5] Mechanical Properties of C3N Nanotubes from Molecular Dynamics Simulation Studies
[6] Calorimetric analysis and molecular dynamics simulation of cure kinetics of epoxy/chitosan-modified Fe3O4 nanocomposites
- This work is resulted from a collaboration among several researchers all over the world. It would be helpful if their contributions are clearly stated.
Response:
- We have mentioned the contribution of authors with more details based on your suggestion that can also be found as follows:
AHM and MRS did conceptualization; MT, AT & AS generated formal analysis; Investigations and data curation were managed by MTM & SH; writing—original draft done by AHM, AS, AT & MT; writing—review and editing by FJS & MRS to give the review its final form; visualization and final checking of the whole manuscript by MRS.
- Many figures in this paper are copied from literature. Have the authors gotten permission to use them from a copyright perspective?
Response:
- Yes, of course. All the used figures have copyright permission. We submitted together with main file the permission assignments as well.
- Simply mistakes can be found throughout this paper, and these should be avoided whatsoever.
- We carefully rechecked the manuscript, and the context is double-checked.
Reviewer 2 Report
Dear authors
There is a very interesting topic in your paper, but in my opinion works seems to be revised, just in order to help readers to better understand .
In general, this article seems more a review that an original research input and there are some concepts as, adsorption, diffusion, permeation that should be differentiate and explained. Using references, there a re no discrimination between systems where authors use MOF's inside a polymeric substrate, or not. This is a very important fact that should be taken into account when talking about mass-transfer phenomena.
In more detailed way,there are some comments to be done:
- Fig 1: In my opinion, it should be divided into two different figures and both, require more explanations about how every energy supplier alter the formation of MOFS. Examples used would be interesting to be divided in the "in situ" formation in substrates (membranes examples) or, in suspension/emulsion systems
- Page 3. Lines 58-59. Please, consider to revise the whole phrase
- Page 3. Line 72. The word "arising" is on different format, please consider to revise
- Page 3. Line 82. LAMMPS is just the acronym of the whole name of software package. Please, consider to include the meaning of this acronym somewhere
- Page 4. Line 87. The word "adsorption" is in different format, please, consider to revise
- In former parts of text, you refer to Higher gradient of Mean Square Displacement, without doing any explanation about the restrictions and condition that this approach requires to get accurate enough results. Please consider to add some details about the theory used here. In my opinion, the references used in this case, are not enough
- Table 1, gives to confusion when using, indiscriminately mass-transport concepts that are very different, without any kind of explanation. Not all readers will be familiar with these complex concepts
- Page 5.Line 102. When talking about abnormal diffusion process, this is not clarified respect adsorption/diffusion concepts. Which is the importance to have, in this case, abnormal diffusion??
- Page 6, Line 110: "suppliers" seems to be in different format, please consider to revise.
Many thanks
Author Response
Authors’ Response to the Review Comments
Journal: Journal of Composite Science
Manuscript #: jcs-831696
Title of Paper: Metal-organic Framework (MOF) through the Lens of Molecular Dynamics Simulation: Current Status and Future Perspective
Authors: Amin Hamed Mashhadzadeh, Ali Taghizadeh, Mohsen Taghizadeh, Muhammad Tajammal Munir, Sajjad Habibzadeh, Azam Salmankhani, Florian J. Stadler, Mohammad Reza Saeb
Dear Editor,
We appreciate the time and efforts spent by the reviewers in qualification of this manuscript. The manuscript has been carefully revised according to the useful comments and suggestions of the reviewers. A point-by-point response to the reviewers’ comments has also been provided. In the revised manuscript we used different colours so as to have attention of reviewers making them able to easily track changes applied to the text. We hope that the revised manuscript can lure the attention of reviewers and respected editor as well.
Thanks for your kind consideration in advance.
Yours sincerely,
Dr. Amin Hamed Mashhadzadeh
Prof. Florian J. Stadler
Dr. Mohammad Reza Saeb
Reviewers' comments:
Reviewer #2:
There is a very interesting topic in your paper, but in my opinion works seems to be revised, just in order to help readers to better understand .
In general, this article seems more a review that an original research input and there are some concepts as, adsorption, diffusion, permeation that should be differentiate and explained. Using references, there are no discrimination between systems where authors use MOF's inside a polymeric substrate, or not. This is a very important fact that should be taken into account when talking about mass-transfer phenomena.
In more detailed way, there are some comments to be done:
Fig 1: In my opinion, it should be divided into two different figures and both, require more explanations about how every energy supplier alter the formation of MOFS. Examples used would be interesting to be divided in the "in situ" formation in substrates (membranes examples) or, in suspension/emulsion systems
Response:
- Authors appreciate your kind suggestion and made changes accordingly. As per your request, we separated Fig. 1 into two figures, and we developed the caption of the figures as well as explanations in context. All changes can be found in blue.
Page 3. Lines 58-59. Please, consider to revise the whole phrase
Response:
- The mentioned paragraph has been revised based on your suggestion and the changes can be fund in blue color.
Page 3. Line 72. The word "arising" is on different format, please consider to revise.
Response:
- We have repaced “arising” with “are originated” as can also be observed in the revised version of manuscript.
Page 3. Line 82. LAMMPS is just the acronym of the whole name of software package. Please, consider to include the meaning of this acronym somewhere.
Response:
- Well noted. It’s been added to the revised manuscript.
Page 4. Line 87. The word "adsorption" is in different format, please, consider to revise
Response:
- It has been corrected.
In former parts of text, you refer to Higher gradient of Mean Square Displacement, without doing any explanation about the restrictions and condition that this approach requires to get accurate enough results. Please consider to add some details about the theory used here. In my opinion, the references used in this case, are not enough.
Response:
- Mean square displacement is a measure for the penetration of molecules inside the MOFs cavities and is indeed a measure of diffusivity. This concept has been used in the context to support and insist on the better diffusivity of MIL-101(Cr) compared to other MOFs considered in that article not as a separate response. To avoid making confusions for the reader, authors decided to eliminate this phrase from the manuscript since this action would not disturb the concept of the paragraph at all.
Table 1, gives to confusion when using, indiscriminately mass-transport concepts that are very different, without any kind of explanation. Not all readers will be familiar with these complex concepts
Response:
- Agreed, the term “mass transfer” is confusing. Needed revises have been made to the table as per your request.
Page 5.Line 102. When talking about abnormal diffusion process, this is not clarified respect adsorption/diffusion concepts. Which is the importance to have, in this case, abnormal diffusion??
Response:
- Thanks for your consideration. By referring to the part of the manuscript you mentioned, we found that the reason of abnormal difuusion of neo-pentane in MIL-47(V) have been expressed by authors clearly according to the following sentence “They also showed that this behavior was not correlated with MIL-47(V) flexibility, but was due to the size of the pores confining the movement of neo-pentane molecules inside the MOF channels”. We have modified the sentence to provide a better comprehension of the explanations we made.
Page 6, Line 110: "suppliers" seems to be in different format, please consider to revise.
Response:
- It was corrected.
Round 2
Reviewer 1 Report
The authors have revised the manuscript according to the reviewers' suggestions.
Reviewer 2 Report
Dear authors
Many thanks for taking into account our comments for revision process. I think that the explanations added fulfill with the requirements presented.
Clarification of certain mechanisms involved in the process of formation of MOF's in situ, will help to readers for a better understanding of the work.
Thanks for the efforts and work developed in the revision.